# Coordinate Descent with Bandit Sampling

**Farnood Salehi**[1]                    **Patrick Thiran**[2]

**L. Elisa Celis**[3]
[1,2,3] School of Computer and Communication Sciences
École Polytechnique Fédérale de Lausanne (EPFL)
`firstname.lastname@epfl.ch`

## Abstract

Coordinate descent methods usually minimize a cost function by updating a random decision variable (corresponding to one coordinate) at a time. Ideally, we would update the decision variable that yields the largest decrease in the cost function. However, finding this coordinate would require checking all of them, which would effectively negate the improvement in computational tractability that coordinate descent is intended to afford. To address this, we propose a new adaptive method for selecting a coordinate. First, we find a lower bound on the amount the cost function decreases when a coordinate is updated. We then use a multi-armed bandit algorithm to learn which coordinates result in the largest lower bound by interleaving this learning with conventional coordinate descent updates except that the coordinate is selected proportionately to the expected decrease. We show that our approach improves the convergence of coordinate descent methods both theoretically and experimentally.

## 1 Introduction

Most supervised learning algorithms minimize an empirical risk cost function over a dataset. Designing fast optimization algorithms for these cost functions is crucial, especially as the size of datasets continues to increase. (Regularized) empirical risk cost functions can often be written as

$$F(\boldsymbol{x}) = f(A\boldsymbol{x}) + \sum_{i=1}^{d} g_i(x_i), \tag{1}$$

where $f(\cdot) : \mathbb{R}^n \longrightarrow \mathbb{R}$ is a smooth convex function, $d$ is the number of decision variables (coordinates) on which the cost function is minimized, which are gathered in vector $\boldsymbol{x} \in \mathbb{R}^d$, $g_i(\cdot) : \mathbb{R} \longrightarrow \mathbb{R}$ are convex functions for all $i \in [d]$, and $A \in \mathbb{R}^{n \times d}$ is the data matrix. As a running example, consider Lasso: if $\boldsymbol{Y} \in \mathbb{R}^n$ are the labels, $f(A\boldsymbol{x}) = 1/2n\|\boldsymbol{Y} - A\boldsymbol{x}\|^2$, where $\|\cdot\|$ stands for the Euclidean norm, and $g_i(x_i) = \lambda|x_i|$. When Lasso is minimized, $d$ is the number of features, whereas when the dual of Lasso is minimized, $d$ is the number of datapoints.

The gradient descent method is widely used to minimize (1). However, computing the gradient of the cost function $F(\cdot)$ can be computationally prohibitive. To bypass this problem, two approaches have been developed: (i) Stochastic Gradient Descent (SGD) selects one *datapoint* to compute an unbiased estimator for the gradient at each time step, and (ii) Coordinate Descent (CD) selects one *coordinate* to optimize over at each time step. In this paper, we focus on improving the latter technique.

When CD was first introduced, algorithms did not differentiate between coordinates; each coordinate $i \in [d]$ was selected uniformly at random at each time step (see, e.g., [19, 20]). However, recent works (see, e.g., [10, 24, 15]) have shown that exploiting the structure of the data and sampling the

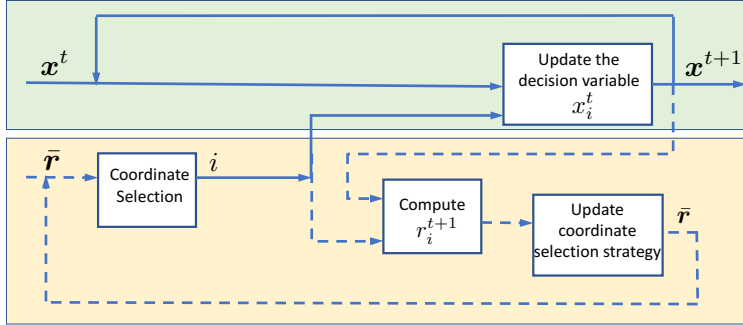

Figure 1: Our approach for coordinate descent. The top (green) part of the approach handles the updates to the decision variable $x_i^t$ (using whichever CD update is desired; our theoretical results hold for updates in the class $\mathcal{H}$ in Definition 4 in the supplementary materials. The bottom (yellow) part of the approach handles the selection of $i \in [d]$ according to a coordinate selection strategy which is updated via bandit optimization (using whichever bandit algorithm is desired) from $r_i^{t+1}$.

coordinates from an appropriate non-uniform distribution can result in better convergence guarantees, both in theory and practice. The challenge is to find the appropriate non-uniform sampling distribution with a lightweight mechanism that maintains the computational tractability of CD.

In this work, we propose a novel adaptive non-uniform coordinate selection method that can be applied to both the primal and dual forms of a cost function. The method exploits the structure of the data to optimize the model by finding and frequently updating the most predictive decision variables. In particular, for each $i \in [d]$ at time $t$, a lower bound $r_i^t$ is derived (which we call the *marginal decrease*) on the amount by which the cost function will decrease when only the $i^{th}$ coordinate is updated.

The marginal decrease $r_i^t$ quantifies by how much updating the $i^{th}$ coordinate is guaranteed to improve the model. The coordinate $i$ with the largest $r_i^t$ is then the one that is updated by the algorithm max_r, described in Section 2.3. This approach is particularly beneficial when the distribution of $r_i^t$s has a high variance across $i$; in such cases updating different coordinates can yield very different decreases in the cost function. For example, if the distribution of $r_i^t$s has a high variance across $i$, max_r is up to $d^2$ times better than uniform sampling, whereas state-of-the-art methods can be at most $d^{3/2}$ better than uniform sampling in such cases (see Theorem 2 in Section 2.3). More precisely, in max_r the convergence speed is proportional to the ratio of the duality gap to the maximum coordinate-wise duality gap. max_r is able to outperform existing adaptive methods because it explicitly finds the coordinates that yield a large decrease of the cost function, instead of computing a distribution over coordinates based on an approximation of the marginal decreases.

However, the computation of the marginal decrease $r_i^t$ for all $i \in [d]$ may still be computationally prohibitive. To bypass this obstacle, we adopt in Section 2.4 a principled approach (B_max_r) for learning the best $r_i^t$s, instead of explicitly computing all of them: At each time $t$, we choose a single coordinate $i$ and update it. Next, we compute the marginal decrease $r_i^t$ of the selected coordinate $i$ and use it as feedback to adapt our coordinate selection strategy using a bandit framework. Thus, in effect, we learn estimates of the $r_i^t$s and simultaneously optimize the cost function (see Figure 1). We prove that this approach can perform almost as well as max_r, yet decreases the number of calculations required by a factor of $d$ (see Proposition 2).

We test this approach on several standard datasets, using different cost functions (including Lasso, logistic and ridge regression) and for both the adaptive setting (the first approach) and the bandit setting (the second approach). We observe that the bandit coordinate selection approach accelerates the convergence of a variety of CD methods (e.g., StingyCD [11] for Lasso in Figure 2, dual CD [18] for $L_1$-regularized logistic-regression in Figure 3, and dual CD [13] for ridge-regression in Figure 3). Furthermore, we observe that in most of the experiments B_max_r (the second approach) converges as fast as max_r (the first approach), while it has the same computational complexity as CD with uniform sampling (see Section 4).

## 2 Technical Contributions

### 2.1 Preliminaries

Consider the following primal-dual optimization pairs

$$\min_{\boldsymbol{x}\in\mathbb{R}^d} F(\boldsymbol{x}) = f(A\boldsymbol{x}) + \sum_{i=1}^{d} g_i(x_i), \quad \min_{\boldsymbol{w}\in\mathbb{R}^n} F_D(\boldsymbol{w}) = f^\star(\boldsymbol{w}) + \sum_{i=1}^{d} g_i^\star(-\boldsymbol{a}_i^\top \boldsymbol{w}), \quad (2)$$

where $A = [\boldsymbol{a}_1, \dots, \boldsymbol{a}_d]$, $\boldsymbol{a}_i \in \mathbb{R}^n$, and $f^\star$ and $g_i^\star$ are the convex conjugates of $f$ and $g_i$, respectively.[1] The goal is to find $\bar{\boldsymbol{x}} := \operatorname{argmin}_{\boldsymbol{x}\in\mathbb{R}^d} F(\boldsymbol{x})$. In rest of the paper, we will need the following notations. We denote by $\epsilon(\boldsymbol{x}) = F(\boldsymbol{x}) - F(\bar{\boldsymbol{x}})$ the *sub-optimality gap* of $F(\boldsymbol{x})$, and by $G(\boldsymbol{x}, \boldsymbol{w}) = F(\boldsymbol{x}) - (-F_D(\boldsymbol{w}))$ the *duality gap* between the primal and the dual solutions, which is an upper bound on $\epsilon(\boldsymbol{x})$ for all $\boldsymbol{x} \in \mathbb{R}^d$. We further use the shorthand $G(\boldsymbol{x})$ for $G(\boldsymbol{x}, \boldsymbol{w})$ when $\boldsymbol{w} = \nabla f(A\boldsymbol{x})$. For $\boldsymbol{w} = \nabla f(A\boldsymbol{x})$, using the Fenchel-Young property $f(A\boldsymbol{x}) + f^\star(\boldsymbol{w}) = (A\boldsymbol{x})^\top \boldsymbol{w}$, $G(\boldsymbol{x})$ can be written as $G(\boldsymbol{x}) = \sum_{i=1}^{d} G_i(\boldsymbol{x})$ where $G_i(\boldsymbol{x}) = \left(g_i^\star(-\boldsymbol{a}_i^\top \boldsymbol{w}) + g_i(x_i) + x_i \boldsymbol{a}_i^\top \boldsymbol{w}\right)$ is the $i^{th}$ *coordinate-wise duality gap*. Finally, we denote by $\kappa_i = \bar{u} - x_i$ the $i^{th}$ *dual residue* where $\bar{u} = \arg\min_{u\in\partial g_i^\star(-\boldsymbol{a}_i^\top \boldsymbol{w})} |u - x_i|$ with $\boldsymbol{w} = \nabla f(A\boldsymbol{x})$.

### 2.2 Marginal Decreases

Our coordinate selection approach works for a class $\mathcal{H}$ of update rules for the decision variable $x_i$. For the ease of exposition, we defer the formal definition of the class $\mathcal{H}$ (Definition 4) to the supplementary materials and give here an informal but more insightful definition. The class $\mathcal{H}$ uses the following *reference* update rule for $x_i$, when $f(\cdot)$ is $1/\beta$-smooth and $g_i$ is $\mu_i$-strongly convex: $x_i^{t+1} = x_i^t + s_i^t \kappa_i^t$, where

$$s_i^t = \min\left\{1, \frac{G_i^t + \mu_i |\kappa_i^t|^2/2}{|\kappa_i^t|^2(\mu_i + \|\boldsymbol{a}_i\|^2/\beta)}\right\}. \quad (3)$$

$\kappa_i^t$, the $i^{th}$ dual residue at time $t$, and $G_i^t$, the $i^{th}$ coordinate-wise duality gap at time $t$, quantify the sub-optimality along coordinate $i$. Because of (3), the effect of $s_i^t$ is to increase the step size of the update of $x_i^t$ when $G_i^t$ is large. The class $\mathcal{H}$ contains also all update rules that decrease the cost function faster than the reference update rule (see two criteria (11) and (12) in Definition 4 in the supplementary materials. For example, the update rules in [18] and [11] for Lasso, the update rules in [20] for hinge-loss SVM and ridge regression, the update rule in [6] for the strongly convex functions, in addition to the reference update rule defined above, belong to this class $\mathcal{H}$.

We begin our analysis with a lemma that provides the marginal decrease $r_i^t$ of updating a coordinate $i \in [d]$ according to any update rule in the class $\mathcal{H}$.

**Lemma 1** *In (1), let $f$ be $1/\beta$-smooth and each $g_i$ be $\mu_i$-strongly convex with convexity parameter $\mu_i \geq 0 \; \forall i \in [d]$. For $\mu_i = 0$, we assume that $g_i$ has a $L$-bounded support. After selecting the coordinate $i \in [d]$ and updating $x_i^t$ with an update rule in $\mathcal{H}$, we have the following guarantee:*

$$F(\boldsymbol{x}^{t+1}) \leq F(\boldsymbol{x}^t) - r_i^t, \quad (4)$$

*where*

$$r_i^t = \begin{cases} G_i^t - \frac{\|\boldsymbol{a}_i\|^2 |\kappa_i^t|^2}{2\beta} & \text{if } s_i^t = 1, \\ \frac{s_i^t \left(G_i^t + \mu_i |\kappa_i^t|^2/2\right)}{2} & \text{otherwise.} \end{cases} \quad (5)$$

In the proof of Lemma 1 in the supplementary materials, the decrease of the cost function is upper-bounded using the smoothness property of $f(\cdot)$ and the convexity of $g_i(\cdot)$ for any update rule in the class $\mathcal{H}$.

**Remark 1** *In the well-known SGD, the cost function $F(\boldsymbol{x}^t)$ might increase at some iterations $t$. In contrast, if we use CD with an update rule in $\mathcal{H}$, it follows from (5) and (3) that $r_i^t \geq 0$ for all $t$, and from (4) that the cost function $F(\boldsymbol{x}^t)$ never increases. This property provides a strong stability guarantee, and explains (in part) the good performance observed in the experiments in Section 4.*

## 2.3 Greedy Algorithms (Full Information Setting)

In first setting, which we call full information setting, we assume that we have computed $r_i^t$ for all $i \in [d]$ and all $t$ (we will relax this assumption in Section 2.4). Our first algorithm max_r makes then a greedy use of Lemma 1, by simply choosing at time $t$ the coordinate $i$ with the largest $r_i^t$.

**Proposition 1 (max_r)** *Under the assumptions of Lemma 1, the optimal coordinate $i_t$ for minimizing the right-hand side of (4) at time $t$ is $i_t = \arg\max_{j \in [d]} r_j^t$.*

**Remark 2** *This rule can be seen as an extension of the Gauss-Southwell rule [13] for the class of cost functions that the gradient does not exist, which selects the coordinate whose gradient has the largest magnitude (when $\nabla_i F(\boldsymbol{x})$ exits), i.e., $i_t = \arg\max_{i \in [d]} |\nabla_i F(\boldsymbol{x})|$. Indeed, Lemma 2 in the supplementary materials shows that for the particular case of $L_2$-regularized cost functions $F(\boldsymbol{x})$, the Gauss-Southwell rule and max_r are equivalent.*

If functions $g_i(\cdot)$ are strongly convex (i.e., $\mu_i > 0$), then max_r results in a linear convergence rate and matches the lower bound in [2].

**Theorem 1** *Let $g_i$ in (1) be $\mu_i$-strongly convex with $\mu_i > 0$ for all $i \in [d]$. Under the assumptions of Lemma 1, we have the following linear convergence guarantee:*

$$\epsilon(\boldsymbol{x}^t) \leq \epsilon(\boldsymbol{x}^0) \prod_{l=1}^{t} \left( 1 - \max_{i \in [d]} \frac{G_i(\boldsymbol{x}^t)\mu_i}{G(\boldsymbol{x}^t)\left(\mu_i + \frac{\|\boldsymbol{a}_i\|^2}{\beta}\right)} \right), \tag{6}$$

*for all $t > 0$, where $\epsilon(\boldsymbol{x}^0)$ is the sub-optimality gap at $t = 0$.*

Now, if functions $g_i(\cdot)$ are not necessary strongly convex (i.e., $\mu_i = 0$), max_r is also very effective and outperforms the state-of-the-art.

**Theorem 2** *Under the assumptions of Lemma 1, let $\mu_i \geq 0$ for all $i \in [d]$. Then,*

$$\epsilon(\boldsymbol{x}^t) \leq \frac{8L^2\eta^2/\beta}{2d + t - t_0} \tag{7}$$

*for all $t \geq t_0$, where $t_0 = \max\{1, 2d \log{}^{d\beta\epsilon(\boldsymbol{x}^0)}/_{4L^2\eta^2}\}$, $\epsilon(\boldsymbol{x}^0)$ is the sub-optimality gap at $t = 0$ and $\eta = O(d)$ is an upper bound on $\min_{i \in [d]} {}^{G(\boldsymbol{x}^t)\,\|\boldsymbol{a}_i\|}/_{G_i(\boldsymbol{x}^t)}$ for all iterations $l \in [t]$.*

To make the convergence bounds (6) and (7) easier to understand, assume that $\mu_i = \mu_1$ and that the data is normalized, so that $\|\boldsymbol{a}_i\| = 1$ for all $i \in [d]$. First, by letting $\eta = O(d)$ be an upper bound on $\min_{i \in [d]} {}^{G(\boldsymbol{x}^t)}/_{G_i(\boldsymbol{x}^t)}$ for all iterations $l \in [t]$, Theorem 1 results in a linear convergence rate, i.e., $\epsilon(\boldsymbol{x}^t) = O\left(\exp(-c_1 t/\eta)\right)$ for some constant $c_1 > 0$ that depends on $\mu_1$ and $\beta$, whereas Theorem 2 provides a sublinear convergence guarantee, i.e., $\epsilon(\boldsymbol{x}^t) = O\left(\eta^2/t\right)$.

Second, note that in both convergence guarantees, we would like to have a small $\eta$. The ratio $\eta$ can be as large as $d$, when the different coordinate-wise gaps $G_i(\boldsymbol{x}^t)$ are equal. In this case, non-uniform sampling does not bring any advantage over uniform sampling, as expected. In contrast, if for instance $c \cdot G(\boldsymbol{x}^t) \leq \max_{i \in [d]} G_i(\boldsymbol{x}^t)$ for some constant $1/d \leq c \leq 1$, then choosing the coordinate with the largest $r_i^t$ results in a decrease in the cost function, that is $1 \leq c \cdot d$ times larger compared to uniform sampling. Theorems 1 and 2 are proven in the supplementary materials.

Finally, let us compare the bound of max_r given in Theorem 2 with the state-of-the-art bounds of ada_gap in Theorem 3.7 of [15] and of CD algorithm in Theorem 2 of [8]. For the sake of simplicity, assume that $\|\boldsymbol{a}_i\| = 1$ for all $i \in [d]$. When $c \cdot G(\boldsymbol{x}^t) \leq \max_{i \in [d]} G_i(\boldsymbol{x}^t)$ and some constant $1/d \leq c \leq 1$, the convergence guarantee for ada_gap is $\mathbb{E}\left[\epsilon(\boldsymbol{x}^t)\right] = O\left(\sqrt{d}L^2/_{\beta(c^2 + 1/d)^{3/2}(2d+t)}\right)$ and the convergence guarantee of the CD algorithm in [8] is $\mathbb{E}\left[\epsilon(\boldsymbol{x}^t)\right] = O\left(dL^2/_{\beta c(2d+t)}\right)$, which are much tighter than the convergence guarantee of CD with uniform sampling $\mathbb{E}\left[\epsilon(\boldsymbol{x}^t)\right] = O\left(d^2 L^2/_{\beta(2d+t)}\right)$. In contrast, the convergence guarantee of max_r is $\epsilon(\boldsymbol{x}^t) = O\left(L^2/_{\beta c^2(2d+t)}\right)$, which is $\sqrt{d}/c$ times better than ada_gap, $dc$ times better than the CD algorithm in [8] and $c^2 d^2$ times better than uniform sampling for the same constant $c \geq 1/d$.

**Remark 3** *There is no randomness in the selection rule used in max_r (beyond tie breaking), hence the convergence results given in Theorems 1 and 2 a.s. hold for all $t$.*

## 2.4 Bandit Algorithms (Partial Information Setting)

State-of-the-art algorithms and max_r require knowing a sub-optimality metric (e.g., $G_i^t$ in [15, 8], the norm of gradient $\nabla_i F(\boldsymbol{x}^t)$ in [13], the marginal decreases $r_i^t$ in this work) for all coordinates $i \in [d]$, which can be computationally expensive if the number of coordinates $d$ is large. To overcome this problem, we use a novel approach inspired by the bandit framework that *learns* the best coordinates over time from the partial information it receives during the training.

**Multi-armed Bandit:** In a multi-armed bandit (MAB) problem, there are $d$ possible arms (which are here the coordinates) that a bandit algorithm can choose from for a reward ($r_i^t$ in this work) at time $t$. The goal of the MAB is to maximize the cumulative rewards that it receives over $T$ rounds (i.e., $\sum_{i=1}^{T} r_{i_t}^t$, where $i_t$ is the arm (coordinate) selected at time $t$). After each round, the MAB only observes the reward of the selected arm $i_t$, and hence has only access to *partial information*, which it then uses to refine its arm (coordinate) selection strategy for the next round.

---

**Algorithm 1** B_max_r

> **input:** $x^0$, $\varepsilon$ and $E$
> **initialize:** set $\bar{r}_i^0 = r_i^0$ for all $i \in [d]$
> **for** $t = 1$ **to** $T$ **do**
>   **if** $t \mod E == 0$ **then**
>     set $\bar{r}_i^t = r_i^t$ for all $i \in [d]$
>   **end if**
>   Generate $K \sim Bern(\varepsilon)$
>   **if** $K == 1$ **then**
>     Select $i_t \in [d]$ uniformly at random
>   **else**
>     Select $i_t = \arg\max_{i \in [d]} \bar{r}_i^t$
>   **end if**
>   Update $x_{i_t}^t$ by an update rule in $\mathcal{H}$
>   Set $\bar{r}_{i_t}^{t+1} = r_{i_t}^{t+1}$ and $\bar{r}_i^{t+1} = \bar{r}_i^t$ for all $i \neq i_t$
> **end for**

---

In our second algorithm B_max_r, the marginal decreases $r_i^t$ computed for all $i \in [d]$ at each round $t$ by max_r are replaced by estimates $\bar{r}_i$ computed by an MAB as follows. First, time is divided into bins of size $E$. At the beginning of a bin $t_e$, the marginal decreases $r_i^{t_e}$ of all coordinates $i \in [d]$ are computed, and the estimates are set to these values ($\bar{r}_i^t = r_i^{t_e}$ for all $i \in [d]$). At each iteration $t_e \leq t \leq t_e + E$ within that bin, with probability $\varepsilon$ a coordinate $i_t \in [d]$ is selected uniformly at random, and otherwise (with probability $(1 - \varepsilon)$) the coordinate with the largest $\bar{r}_i^t$ is selected. Coordinate $i_t$ is next updated, as well as the estimate of the marginal decrease $\bar{r}_{i_t}^{t+1} = r_{i_t}^{t+1}$, whereas the other estimates $\bar{r}_j^{t+1}$ remain unchanged for $j \neq i_t$. The algorithm can be seen as a modified version of $\varepsilon$-greedy (see [3]) that is devel-

oped for the setting where the reward of arms follow a fixed probability distribution, $\varepsilon$-greedy uses the empirical mean of the observed rewards as an estimate of the rewards. In contrast, in our setting, the rewards do not follow such a fixed probability distribution and the most recently observed reward is the best estimate of the reward that we could have. In B_max_r, we choose $E$ not too large and $\varepsilon$ large enough such that every arm (coordinate) is sampled often enough to maintain an accurate estimate of the rewards $r_i^t$ (we use $E = O(d)$ and $\varepsilon = 1/2$ in the experiments of Section 4).

The next proposition shows the effect of the estimation error on the convergence rate.

**Proposition 2** *Consider the same assumptions as Lemma 1 and Theorem 2. For simplicity, let $\|\boldsymbol{a}_i\| = \|\boldsymbol{a}_1\|$ for all $i \in [d]$ and $\epsilon(\boldsymbol{x}^0) \leq \sqrt{2\alpha L^2 \|\boldsymbol{a}_1\|^2/\beta \left(\varepsilon/d + 1 - \varepsilon/c\right)} = O(d)$.[2] Let $j_\star^t = \arg\max_{i \in [d]} \bar{r}_i^t$. If $\max_{i \in [d]} r_i^t/r_{j_\star^t}^t \leq c(E, \varepsilon)$ for some finite constant $c = c(E, \varepsilon)$, then by using B_max_r (with bin size $E$ and exploration parameter $\varepsilon$) we have*

$$\mathbb{E}\left[\epsilon(\boldsymbol{x}^t)\right] \leq \frac{\alpha}{2 + t - t_0}, \quad \text{where} \quad \alpha = \frac{8L^2 \|\boldsymbol{a}_1\|^2}{\beta \left(\varepsilon/d^2 + (1-\varepsilon)/\eta^2 c\right)}, \tag{8}$$

*for all $t \geq t_0 = \max\left\{1, 4\epsilon(\boldsymbol{x}^0)/\alpha \log(2\epsilon(\boldsymbol{x}^0)/\alpha)\right\} = O(d)$ and where $\eta$ is an upper bound on $\min_{i \in [d]} G(\boldsymbol{x}^t)/G_i(\boldsymbol{x}^t)$ for iterations $l \in [t]$.*

*What is the effect of $c(E, \varepsilon)$?* In Proposition 2, $c = c(E, \varepsilon)$ upper bounds the estimation error of the marginal decreases $r_i^t$. To make the effect of $c(E, \varepsilon)$ on the convergence bound (8) easier to

Table 1: The shaded rows correspond to the algorithms introduced in this paper. $\bar{z}$ denotes the number of non-zero entries of the data matrix $A$. The numbers below the column dataset/cost are the clock time (in seconds) needed for the algorithms to reach a sub-optimality gap of $\epsilon(\boldsymbol{x}^t) = \exp(-5)$.

| method | computational cost (per epoch) | dataset/cost | | |
|---|---|---|---|---|
| | | aloi/Lasso | a9a/log reg | usps/ridge reg |
| uniform | $O(\bar{z})$ | 27.8 | 11.8 | 1 |
| ada_gap | $O(d \cdot \bar{z})$ | 52.8 | 42.4 | 88 |
| max_r | $O(d \cdot \bar{z})$ | 6.2 | 4.5 | 9.5 |
| gap_per_epoch | $O(\bar{z} + d \log d)$ | 75 | 11.1 | 300 |
| Approx | $O(\bar{z} + d \log d)$ | 16.3 | 2.3 | - |
| NUACDM | $O(\bar{z} + d \log d)$ | - | - | 6 |
| B_max_r | $O(\bar{z} + d \log d)$ | 11 | 1.9 | 1 |

understand, let $\varepsilon = 1/2$, then $\alpha \sim 1/(1/d^2 + 1/\eta^2 c)$. We can see from the convergence bound (8) and the value of $\alpha$ that if $c$ is large, the convergence rate is proportional to $d^2$ similarly to uniform sampling (i.e., $\epsilon(\boldsymbol{x}^t) \in O(d^2/t)$). Otherwise, if $c$ is small, the convergence rate is similar to max_r ($\epsilon(\boldsymbol{x}^t) \in O(\eta^2/t)$, see Theorem 2).

*How to control $c = c(E, \varepsilon)$?* We can control the value of $c$ by varying the bin size $E$. Doing so, there is a trade-off between the value of $c$ and the average computational cost of an iteration. On the one hand, if we set the bin size to $E = 1$ (i.e., full information setting), then $c = 1$ and B_max_r boils down to max_r, while the average computational cost of an iteration is $O(nd)$. On the other hand, if $E > 1$ (i.e., partial information setting), then $c \geq 1$, while the average computational complexity of an iteration is $O(nd/E)$. In our experiments, we find that by setting $d/2 \leq E \leq d$, B_max_r converges faster than uniform sampling (and other state-of-the-art methods) while the average computational cost of an iteration is $O(n + \log d)$, similarly to the computational cost of an iteration of CD with uniform sampling ($O(n)$), see Figures 2 and 3. We also find that any exploration parameter $\varepsilon \in [0.2, 0.7]$ in B_max_r works reasonably well. The proof of Proposition 2 is similar to the proof of Theorem 2 and is given in the supplementary materials.

## 3 Related Work

Non-uniform coordinate selection has been proposed first for constant (non-adaptive) probability distributions $p$ over $[d]$. In [24], $p_i$ is proportional to the Lipschitz constant of $g_i^\star$. Similar distributions are used in [1, 23] for strongly convex $f$ in (1).

Time varying (adaptive) distributions, such as $p_i^t = |\kappa_i^t|/(\sum_{j=1}^d |\kappa_j^t|)$ [6], and $p_i^t = G_i(\boldsymbol{x}^t)/G(\boldsymbol{x}^t)$ [15, 14], have also been considered. In all these cases, the full information setting is used, which requires the computation of the distribution $p^t$ ($\Omega(nd)$ calculations) at each step. To bypass this problem, heuristics are often used; e.g., $p^t$ is calculated once at the beginning of an epoch of length $E$ and is left unchanged throughout the remainder of that epoch. This heuristic approach does not work well in a scenario where $G_i(\boldsymbol{x}^t)$ varies significantly. In [8] a similar idea to max_r is used with $r_i$ replaced by $G_i$, but only in the full information setting. Because of the update rule used in [8], the convergence rate is $O(d \cdot \max G_i(\boldsymbol{x}^t)/G(\boldsymbol{x}^t))$ times slower than Theorem 2 (see also the comparison at the end of Section 2.3). The Gauss-Southwell rule (GS) is another coordinate selection strategy for smooth cost functions [21] and its convergence is studied in [13] and [22]. GS selects the coordinate to update as the one that maximizes $|\nabla_i F(\boldsymbol{x}^t)|$ at time $t$. max_r can be seen as an extension of GS to a broader class of cost functions (see Lemma 2 in the supplementary materials). Furthermore, when only sub-gradients are defined for $g_i(\cdot)$, GS needs to solve a proximal problem. To address the computational tractability of GS, in [22], lower and upper bounds on the gradients are computed (instead of computing the gradient itself) and used for selecting the coordinates, but these lower and upper bounds might be loose and/or difficult to find. For example, without a heavy pre-processing of the data, ASCD in [22] converges with the same rate as uniform sampling when the data is normalized and $f(A\boldsymbol{x}) = \|A\boldsymbol{x} - \boldsymbol{Y}\|^2$.

In contrast, our principled approach leverages a bandit algorithm to learn a good estimate of $r_i^t$; this allows for theoretical guarantees and outperforms the state-of-the-art methods, as we will see in Section 4. Furthermore, our approach does not require the cost function to be strongly convex (contrary to e.g., [6, 13])

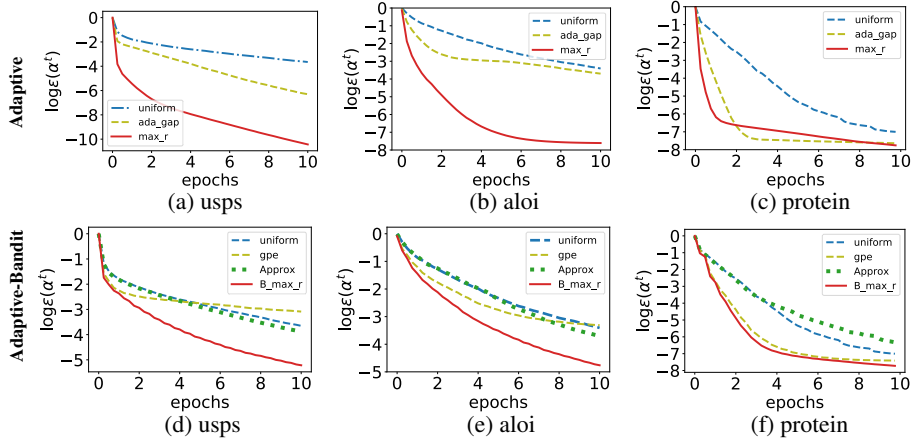

Figure 2: CD for regression using Lasso (i.e., a non-smooth cost function). Y-axis is the log of sub-optimality gap and x-axis is the number of epochs. The algorithms presented in this paper (max_r, B_max_r) outperform the state-of-the-art across the board.

Bandit approaches have very recently been used to accelerate various stochastic optimization algorithms; among these works [12, 17, 16, 4] focus on improving the convergence of SGD by reducing the variance of the estimator for the gradient. A bandit approach is also used in [12] to sample for CD. However, instead of using the bandit to minimize the cost function directly as in B_max_r, it is used to minimize the variance of the estimated gradient. This results in a $O(1/\sqrt{t})$ convergence, whereas the approach in our paper attains an $O(1/t)$ rate of convergence. In [16] bandits are used to find the coordinate $i$ whose gradient has the largest magnitude (similar to GS). At each round $t$ a stochastic bandit problem is solved from scratch, ignoring all past information prior to $t$, which, depending on the number of datapoints, might require many iterations. In contrast, our method incorporates past information and needs only one sample per iteration.

## 4 Empirical Simulations

We compare the algorithms from this paper with the state-of-the-art approaches, in two ways. First, we compare the algorithm (max_r) for full information setting as in Section 2.3 against other state-of-the-art methods that similarly use $O(d \cdot \bar{z})$ computations per epoch of size $d$, where $\bar{z}$ denotes the number of non-zero elements of $A$. Next, we compare the algorithm for partial information setting as in Section 2.4 (B_max_r) against other methods with appropriate heuristic modifications that also allow them to use $O(\bar{z})$ computations per epoch. The datasets we use are found in [5]; we consider usps, aloi and protein for regression, and w8a and a9a for binary classification (see Table 2 in the supplementary materials for statistics about these datasets).

Various cost functions are considered for the experiments, including a strongly convex cost function (ridge regression) and non-smooth cost functions (Lasso and $L_1$-regularized logistic regression). These cost functions are optimized using different algorithms, which minimize either the primal or the dual cost function. The convergence time is the metric that we use to evaluate different algorithms.

### 4.1 Experimental Setup

Benchmarks for Adaptive Algorithm (max_r):

- **uniform** [18]: Sample a coordinate $i \in [n]$ uniformly at random.[3]
- **ada_gap** [15]: Sample a coordinate $i \in [n]$ with probability $G_i(\boldsymbol{x}^t)/G(\boldsymbol{x}^t)$.

Benchmarks for Adaptive-Bandit Algorithm (B_max_r): For comparison, in addition to the uniform sampling, we consider the coordinate selection method that has the best performance empirically in [15] and two accelerated CD methods NUACDM in [1] and Approx in [9].

- **gpe** [15]: This algorithm is a heuristic version of ada_gap, where the sampling probability $p_i^t = G_i(\boldsymbol{x}^t)/G(\boldsymbol{x}^t)$ for $i \in [d]$ is re-computed once at the beginning of each bin of length $E$.

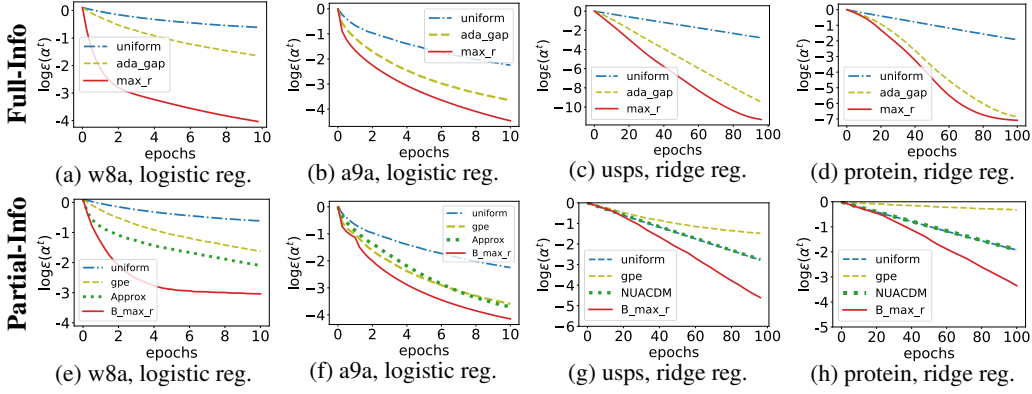

Figure 3: CD for binary Classification using $L_1$-regularized logistic regression and CD for regression using Lasso. The algorithms presented in this paper (max_r and B_max_r) outperform the state-of-the-art across the board.

- **NUACDM** [1]: Sample a coordinate $i \in [d]$ with probability proportional to the square root of smoothness of the cost function along the $i^{th}$ coordinate, then use an unbiased estimator for the gradient to update the decision variables.
- **Approx** [9]: Sample a coordinate $i \in [d]$ uniformly at random, then use an unbiased estimator for the gradient to update the decision variables.

NUACDM is the state-of-the-art accelerated CD method (see Figures 2 and 3 in [1]) for smooth cost functions. Approx is an accelerated CD method proposed for cost functions with non-smooth $g_i(\cdot)$ in ([1]). We implemented Approx for such cost functions in Lasso and $L_1$-regularized logistic regression. We also implemented Approx for ridge-regression but NUACDM converged faster in our setting, whereas for the smoothen version of Lasso but Approx converged faster than NUACDM in our setting. The origin of the computational cost is two-fold: Sampling a coordinate $i$ and updating it. The average computational cost of the algorithms for $E = d/2$ is depicted in Table 1. Next, we explain the setups and update rules used in the experiments.

For Lasso $F(\boldsymbol{x}) = 1/2n\|\boldsymbol{Y} - A\boldsymbol{x}\|^2 + \sum_{i=1}^n \lambda|x_i|$. We consider the stingyCD update proposed in [11]: $x_i^{t+1} = \arg\min_z \left[f(A\boldsymbol{x}^t + (z - x_i^t)\boldsymbol{a}_i)\right] + g_i(x_i)$. In Lasso, the $g_i$s are not strongly convex ($\mu_i = 0$). Therefore, for computing the dual residue, the Lipschitzing technique in [7] is used, i.e., $g_i(\cdot)$ is assumed to have bounded support of size $B = F(\boldsymbol{x}^0)/\lambda$ and $g_i^\star(u_i) = B \max\{|u_i| - \lambda, 0\}$.

For logistic regression $F(\boldsymbol{x}) = 1/n \sum_{i=1}^n \log\left(1 + \exp(-y_i \cdot \boldsymbol{x}^\top \boldsymbol{a}_i)\right) + \sum_{i=1}^n \lambda|x_i|$. We consider the update rule proposed in [18]: $x_i^{t+1} = s_{4\lambda}(x_i^t - 4\partial f(A\boldsymbol{x}^t)/\partial x_i)$, where $s_\lambda(q) = \text{sign}(q) \max\{|q| - \lambda, 0\}$.

For ridge regression $F(\boldsymbol{x}) = 1/n\|\boldsymbol{Y} - A\boldsymbol{x}\|^2 + \lambda/2\|\boldsymbol{x}\|^2$ and it is strongly convex. We consider the update proposed for the dual of ridge regression in [20], hence B_max_r and other adaptive methods select one of the dual decision variables to update.

In all experiments, $\lambda$s are chosen such that the test and train errors are comparable, and all update rules belong to $\mathcal{H}$. In addition, in all experiments, $E = d/2$ in B_max_r and gap_per_epoch. Recall that when minimizing the primal, $d$ is the number of features and when minimizing the dual, $d$ is the number of datapoints.

## 4.2 Empirical Results

Figure 2 shows the result for Lasso. Among the adaptive algorithms, max_r outperforms the state-of-the-art (see Figures 2a, 2b and 2c). Among the adaptive-bandit algorithms, B_max_r outperforms the benchmarks (see Figures 2d, 2e and 2f). We also see that B_max_r converges slower than max_r for the same number of iterations, but we note that an iteration of B_max_r is $O(d)$ times cheaper than max_r. For logistic regression, see Figures 3a, 3b, 3e and 3f. Again, those algorithms outperform the state-of-the-art. We also see that B_max_r converges with the same rate as max_r. We see that the accelerated CD method Approx converges faster than uniform sampling and gap_per_epoch, but using B_max_r improves the convergence rate and reaches a lower sub-optimality gap $\epsilon$ with the same number of iterations. For ridge regression, we see in Figures 3c, 3d that max_r converges faster

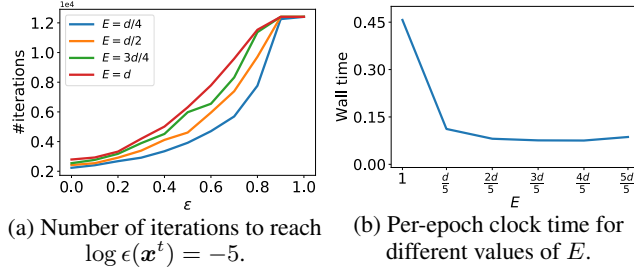

(a) Number of iterations to reach
$\log \epsilon(\boldsymbol{x}^t) = -5$.

(b) Per-epoch clock time for
different values of $E$.

Figure 4: Analysis of the running time of B_max_r for different values of $\varepsilon$ and $E$. A smaller $E$ results in fewer iterations, and results in larger clock time per epoch (an epoch is $d$ iterations of CD).

than the state-of-the-art ada-gap. We also see in Figures 3g, 3h that B_max_r converges faster than other algorithms. gap_per_epoch performs poorly because it is unable to adapt to the variability of the coordinate-wise duality gaps $G_i$ that vary a lot from one iteration to the next. In contrast, this variation slows down the convergence of B_max_r compared to max_r, but B_max_r is still able to cope with this change by exploring and updating the estimations of the marginal decreases. In the experiments we report the sub-optimality gap as a function of the number of iterations, but the results are also favourable when we report them as a function of actual time. To clarify, we compare the clock time needed by each algorithm to reach a sub-optimality gap $\epsilon(\boldsymbol{x}^t) = \exp(-5)$ in Table 1.[4]

Next, we study the choice of parameters $\varepsilon$ and $E$ in Algorithm 1. As explained in Section 2.4 the choice of these two parameters affect $c$ in Proposition 2, hence the convergence rate. To test the effect of $\varepsilon$ and $E$ on the convergence rate, we choose a9a dataset and perform a binary classification on it by using the logistic regression cost function. Figure 4a depicts the number of iterations required to reach the log-suboptimality gap $\log \epsilon$ of $-5$. In the top-right corner, $\varepsilon = 1$ and B_max_r becomes CD with uniform sampling (for any value of $E$). As expected, for any $\varepsilon$, the smaller $E$, the smaller the number of iterations to reach the log-suboptimality gap of $-5$. This means that $c(\varepsilon, E)$ is a decreasing function of $E$. Also, we see that as $\varepsilon$ increases, the convergence becomes slower. That implies that for this dataset and cost function $c(\varepsilon, E)$ is close to 1 for all $\varepsilon$ hence there is no need for exploration and a smaller value for $\varepsilon$ can be chosen. Figure 4b depicts the per epoch clock time for $\varepsilon = 0.5$ and different values of $E$. Note that the clock time is not a function of $\varepsilon$. As expected, a smaller bin size $E$ results in a larger clock time, because we need to compute the marginal decreases for all coordinates more often. After $E = 2d/5$ we see that clock time does not decrease much, this can be explained by the fact that for large enough $E$ computing the gradient takes more clock time than computing the marginal decreases.

## 5   Conclusion

In this work, we propose a new approach to select the coordinates to update in CD methods. We derive a lower bound on the decrease of the cost function in Lemma 1, i.e., the marginal decrease, when a coordinate is updated, for a large class of update methods $\mathcal{H}$. We use the marginal decreases to quantify how much updating a coordinate improves the model. Next, we use a bandit algorithm to *learn* which coordinates decrease the cost function significantly throughout the course of the optimization algorithm by using the marginal decreases as feedback (see Figure 1). We show that the approach converges faster than state-of-the-art approaches both theoretically and empirically. We emphasize that our coordinate selection approach is quite general and works for a large class of update rules $\mathcal{H}$, which includes Lasso, SVM, ridge and logistic regression, and a large class of bandit algorithms that select the coordinate to update.

The bandit algorithm B_max_r uses only the marginal decrease of the selected coordinate to update the estimations of the marginal decreases. An important open question is to understand the effect of having additional budget to choose multiple coordinates at each time $t$. The challenge lies in designing appropriate algorithms to invest this budget to update the coordinate selection strategy such that B_max_r performance becomes even closer to max_r.

## Footnotes

[1]Recall that the convex conjugate of a function $h(\cdot): \mathbb{R}^d \longrightarrow \mathbb{R}$ is $h^\star(\boldsymbol{x}) = \sup_{\boldsymbol{v}\in\mathbb{R}^d}\{\boldsymbol{x}^\top \boldsymbol{v} - h(\boldsymbol{v})\}$.

[2]These assumptions are not necessary but they make the analysis simpler. For example, even if $\epsilon(\boldsymbol{x}^0)$ does not satisfy the required condition, we can scale down $F(\boldsymbol{x})$ by $m$ so that $F(\boldsymbol{x})/m$ is minimized. The new sub-optimality gap becomes $\epsilon(\boldsymbol{x}^0)/m$, and for a sufficiently large $m$ the initial condition is satisfied.

[3] If $\|\boldsymbol{a}_i\| = \|\boldsymbol{a}_j\| \ \forall i, j \in [n]$, importance sampling method in [24] is equivalent to uniform in Lasso and logistic regression.

[4]In our numerical experiments, all algorithms are optimized as much as possible by avoiding any unnecessary computations, by using efficient data structures for sampling, by reusing the computed values from past iterations and (if possible) by writing the computations in efficient matrix form.

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
