[Supplementary Material]

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

# Appendix

## A   Basic Definitions

For completeness, in this section we recall a variety of standard definitions.

### A.1   Basic Definitions

**Definition 1 ($1/\beta$-smooth)** *A function $f(\cdot) : \mathbb{R}^n \longrightarrow \mathbb{R}$ is $1/\beta$-smooth if for any $\boldsymbol{x} \in \mathbb{R}^n$ and $\boldsymbol{y} \in \mathbb{R}^n$*

$$f(\boldsymbol{y}) \leq f(\boldsymbol{x}) + \nabla f(\boldsymbol{x})^\top (\boldsymbol{y} - \boldsymbol{x}) + \frac{1}{2\beta} \|\boldsymbol{x} - \boldsymbol{y}\|^2.$$

**Definition 2 ($\mu$-strongly convex)** *A function $f(\cdot) : \mathbb{R}^n \longrightarrow \mathbb{R}$ is $\mu$-strongly convex if for any $\boldsymbol{x} \in \mathbb{R}^n$ and $\boldsymbol{y} \in \mathbb{R}^n$*

$$f(\boldsymbol{y}) \geq f(\boldsymbol{x}) + \nabla f(\boldsymbol{x})^\top (\boldsymbol{y} - \boldsymbol{x}) + \frac{\mu}{2} \|\boldsymbol{x} - \boldsymbol{y}\|^2.$$

**Definition 3 ($L$-bounded support)** *A function $f(\cdot) : \mathbb{R}^n \longrightarrow \mathbb{R}$ has $L$-bounded support if there exists a euclidean ball with radius $L$ such*

$$f(\boldsymbol{x}) < \infty \Rightarrow \|\boldsymbol{x}\| \leq L.$$

### A.2   The Class of Update Rules

**Definition 4 ($\mathcal{H}$)** *In (1), let $f(\cdot)$ be $1/\beta$-smooth and each $g_i(\cdot)$ be $\mu_i$-strongly convex with convexity parameter $\mu_i \geq 0 \ \forall i \in [d]$. For $\mu_i = 0$, we assume that $g_i$ has a $L_i$-bounded support. Let $\widehat{h} : \mathbb{R}^n \times [n] \longrightarrow \mathbb{R}^n$ be the update rule, i.e., $\boldsymbol{x}^{t+1} = \widehat{h}(\boldsymbol{x}^t, i)$, for the decision variables $\boldsymbol{x}^t$ whose $j^{th}$ entry is*

$$\widehat{h}_j(\boldsymbol{x}, i) = \begin{cases} x_j + s_j \kappa_j & \text{if } j = i, \\ x_j & \text{if } j \neq i, \end{cases} \tag{9}$$

*where*

$$s_i = \min \left\{ 1, \frac{G_i(\boldsymbol{x}) + \mu_i |\kappa_i|^2 / 2}{|\kappa_i|^2 (\mu_i + \|\boldsymbol{a}_i\|^2 / \beta)} \right\}. \tag{10}$$

*We use the update $\widehat{h}$ as a baseline to define $\mathcal{H}$. $\mathcal{H}$ is the class of all update rules $h : \mathbb{R}^n \times [n] \longrightarrow \mathbb{R}^n$ such that $\forall \boldsymbol{x} \in \mathbb{R}^n$ and $i \in [d]$,*

$$F\left(h(\boldsymbol{x}, i)\right) \leq F\left(\widehat{h}(\boldsymbol{x}, i)\right), \tag{11}$$

*or*

$$\widehat{F}_P\left(\boldsymbol{x}, h(\boldsymbol{x}, i)\right) \leq \widehat{F}_P\left(\boldsymbol{x}, \widehat{h}(\boldsymbol{x}, i)\right), \tag{12}$$

*where*

$$\widehat{F}_P(\boldsymbol{x}, \boldsymbol{x}') = \sum_{i=1}^d \left( \left(\nabla f(A\boldsymbol{x})^\top \boldsymbol{a}_i\right)(x_i' - x_i) + \frac{1}{2\beta} \|\boldsymbol{a}_i\|^2 (x_i' - x_i)^2 + g_i(x_i') - g_i(x_i) \right). \tag{13}$$

Intuitively, $\widehat{F}_P(\boldsymbol{x}, \boldsymbol{x}')$ approximates the difference of the cost function evaluated at $\boldsymbol{x}$ and $\boldsymbol{x}'$, which follows from the smoothness property of $f$:

$$F(\boldsymbol{x}') - F(\boldsymbol{x}) \leq \nabla f(A\boldsymbol{x})^\top \left( \sum_{i=1}^d \boldsymbol{a}_i(x_i' - x_i) \right) + \frac{1}{2\beta} \left\| \sum_{i=1}^d \boldsymbol{a}_i(x_i' - x_i) \right\|^2 + \sum_{i=1}^d g_i(x_i') - g_i(x_i)$$

$$\leq \sum_{i=1}^d \left( \left(\nabla f(A\boldsymbol{x})^\top \boldsymbol{a}_i\right)(x_i' - x_i) + \frac{1}{2\beta} \|\boldsymbol{a}_i\|^2 (x_i' - x_i)^2 + g_i(x_i') - g_i(x_i) \right),$$

where the first inequality follows from the smoothness property of $f$ and the last inequality follows from the triangle inequality.

# B Proofs

## B.1 Omitted Proofs for CD (Sections 2.2, 2.3 and 2.4)

In this section, we present the proofs of the results in Sections 2.2, 2.3 and 2.4.

**Proof of Lemma 1:** We first prove the claim for the update rule $\widehat{h}$ given by (9) in part (i), and next extend it to any update rule in $\mathcal{H}$ in part (ii).

(i) Our starting point is the inequality

$$F(\boldsymbol{x}^{t+1}) \leq \ F(\boldsymbol{x}^t) - s_i^t G_i(\boldsymbol{x}^t) - \Big(\frac{\mu_i\left(s_i^t - (s_i^t)^2\right)}{2} - \frac{(s_i^t)^2\|\boldsymbol{a}_i\|^2}{2\beta}\Big)|\kappa_i^t|^2, \qquad (14)$$

which holds for $s_i^t \in [0,1]$, for all $i \in [d]$ and which follows from Lemma 3.1 of [15].[5] After minimizing the right-hand side of (14) with respect to $s_i^t$, we attain the desired bound (4) for $s_i^t$ as in (3).

(ii) We now extend (i) to any update rule in $\mathcal{H}$. If the update rule $h(\boldsymbol{x}^t, i)$ satisfies (11), we can easily recover (4) because

$$F\left(h(\boldsymbol{x}^t, i)\right) \leq F\left(\widehat{h}(\boldsymbol{x}^t, i)\right) \leq F(\boldsymbol{x}^t) - r_i^t.$$

If the update rule satisfies (12), we have

$$F\left(h(\boldsymbol{x}^t, i)\right) \leq F\left(\boldsymbol{x}^t\right) + \widehat{F}_P\left(\boldsymbol{x}^t, h(\boldsymbol{x}^t, i)\right) \qquad (15)$$

$$\leq F\left(\boldsymbol{x}^t\right) + \widehat{F}_P\left(\boldsymbol{x}^t, \widehat{h}(\boldsymbol{x}^t, i)\right) \qquad (16)$$

$$\leq F\left(\boldsymbol{x}^t\right) - r_i^t, \qquad (17)$$

where (15) follows from the $1/\beta$-smoothness of $f$ and (13), (16) follows from (12), and (17) follows from the $\mu_i$-strong convexity of $g_i$. More precisely, by plugging

$$g_i(x_i^t + s_i^t \kappa_i^t) = g_i\left(x_i^t + s_i^t(u^t - x_i^t)\right) \leq$$

$$s_i^t g_i(u^t) + (1 - s_i^t)g_i(x_i^t) - \frac{\mu_i}{2}s_i^t(1 - s_i^t)(\kappa_i^t)^2$$

into (16), and using the Fenchel-Young property, we recover (14). Then, by setting $s_i^t$ as in (3) we recover (17). □

**Lemma 2** *Under the assumptions of Lemma 1, if $g_i(x_i) = \lambda \cdot (x_i)^2$ in (1) and $\|\boldsymbol{a}_i\| = 1$ for all $i \in [d]$, then the Gauss-Southwell rule and max_r are equivalent.*

**Proof:** [Proof of Lemma 2] We prove the lemma for $\boldsymbol{x} \in \mathbb{R}^n$ and drop the dependence on $t$ throughout the proof. First, we show that $G_i \sim (\nabla_i F(\boldsymbol{x}))^2$. The function $g_i(x_i) = \lambda \cdot (x_i)^2$ is $2\lambda$ strongly convex for all $i \in [d]$, i.e., $\mu_i = \mu = 2\lambda$. The dual convex conjugate of the function $g_i(x_i) = \lambda \cdot (x_i)^2$ is

$$g^\star(z) = \frac{z^2}{4\lambda}.$$

Then, for $\boldsymbol{w} = \nabla f(A\boldsymbol{x})$, $G_i(\boldsymbol{x}) = g_i^\star(-\boldsymbol{a}_i^\top \boldsymbol{w}) + g_i(x_i) + x_i \boldsymbol{a}_i^\top \boldsymbol{w}$ becomes

$$G_i(\boldsymbol{x}) = \frac{(\boldsymbol{a}_i^\top \boldsymbol{w})^2}{4\lambda} + \lambda(x_i)^2 + x_i \boldsymbol{a}_i^\top \boldsymbol{w} = \frac{\left(\boldsymbol{a}_i^\top \boldsymbol{w} + 2\lambda x_i\right)^2}{4\lambda}.$$

As $\nabla_i F(\boldsymbol{x}) = \boldsymbol{a}_i^\top \nabla f(A\boldsymbol{x}) + 2\lambda x_i = \boldsymbol{a}_i^\top \boldsymbol{w} + 2\lambda x_i$, we have

$$G_i(\boldsymbol{x}) = \frac{(\nabla_i F(\boldsymbol{x}))^2}{4\lambda}.$$

Next, note that

$$\kappa_i = \partial g_i^\star(-\boldsymbol{a}_i^\top \boldsymbol{w}) - x_i = \frac{-\boldsymbol{a}_i^\top \boldsymbol{w}}{2\lambda} - x_i = -\frac{\nabla_i F(\boldsymbol{x})}{2\lambda}.$$

Next, plugging $G_i(\boldsymbol{x}) = (\nabla_i F(\boldsymbol{x}))^2/4\lambda$ and $\kappa_i = -\nabla_i F(\boldsymbol{x})/2\lambda$ in (3) yields

$$s_i = \min\left\{1, \frac{3\lambda}{2\lambda + \frac{1}{\beta}}\right\} \text{ for all } i \in [d].$$

Hence, $r_i$ in (5) becomes

$$r_i = \begin{cases} \frac{(\nabla_i F(\boldsymbol{x}))^2}{8\lambda^2\beta}(2\lambda\beta - 1) & \text{if } \lambda \geq \frac{1}{\beta}, \\ \frac{3}{4}\frac{(\nabla_i F(\boldsymbol{x}))^2}{2\lambda + \frac{1}{\beta}} & \text{otherwise,} \end{cases}$$

Hence, $\arg\max_{i\in[d]} r_i = \arg\max_{i\in[d]} (\nabla_i F(\boldsymbol{x}))^2$. In Gauss-Southwell rule, we choose the coordinate whose gradient has the largest magnitude, i.e., $\arg\max_{i\in[d]} |\nabla_i F(\boldsymbol{x})|$. As a result, the selection rules max_r and Gauss-Southwell rule are equivalent. $\qquad\square$

Now, we show that our approach leads to a linear convergence when $g_i$ is strongly convex for $i \in [d]$, i.e., when $\mu_i > 0$.

**Proof:** **[Proof of Theorem 1]** According to Proposition 1, we know that the selection rule max_r is optimal for the bound (4). Therefore, if we prove the convergence results using (4) for another selection rule, then the same convergence result holds for max_r. For this proof, we use the following selection rule: At time $l$, we choose the coordinate $i$ with the largest $G_i(\boldsymbol{x}^t)\mu_i/(\mu_i+\|\boldsymbol{a}_i\|^2/\beta)$, which we denote by $i^\star$.

First, we show that $r_{i^\star}^t$ in (5) is lower bounded as follows

$$r_{i^\star}^t \geq G_{i^\star}(\boldsymbol{x}^t)\frac{\mu_{i^\star}}{\mu_{i^\star} + \frac{\|\boldsymbol{a}_{i^\star}\|^2}{\beta}}. \tag{18}$$

We prove (18) for two cases $s_{i^\star}^t = 1$ and $s_{i^\star}^t < 1$ separately, where $s_i^t$ is defined in (3) for $i \in [d]$.

(a) If $s_{i^\star}^t = 1$, according to (5) we have

$$r_{i^\star}^t = G_{i^\star}(\boldsymbol{x}^t) - \frac{\|\boldsymbol{a}_{i^\star}\|^2|\kappa_{i^\star}^t|^2}{2\beta}.$$

Next, we prove (18) by showing that $r_{i^\star}^t - G_{i^\star}(\boldsymbol{x}^t)\frac{\mu_{i^\star}}{\mu_{i^\star}+\|\boldsymbol{a}_{i^\star}\|^2/\beta} \geq 0$,

$$r_{i^\star}^t - G_{i^\star}(\boldsymbol{x}^t)\frac{\mu_{i^\star}}{\mu_{i^\star} + \frac{\|\boldsymbol{a}_{i^\star}\|^2}{\beta}}$$

$$= G_{i^\star}(\boldsymbol{x}^t)\frac{\frac{\|\boldsymbol{a}_{i^\star}\|^2}{\beta}}{\mu_{i^\star} + \frac{\|\boldsymbol{a}_{i^\star}\|^2}{\beta}} - \frac{\|\boldsymbol{a}_{i^\star}\|^2|\kappa_{i^\star}^t|^2}{2\beta}$$

$$= \frac{\|\boldsymbol{a}_{i^\star}\|^2}{2\beta} \cdot \frac{2G_{i^\star}(\boldsymbol{x}^t) - \mu_i|\kappa_{i^\star}^t|^2 - \frac{\|\boldsymbol{a}_{i^\star}\|^2|\kappa_{i^\star}^t|^2}{\beta}}{\mu_{i^\star} + \frac{\|\boldsymbol{a}_{i^\star}\|^2}{\beta}} \geq 0,$$

where the last inequality follows by setting $s_{i^\star}^t = 1$ in (3) which then reads:

$$G_{i^\star}(\boldsymbol{x}^t) - \frac{\mu_{i^\star}|\kappa_{i^\star}^t|^2}{2} - \frac{\|\boldsymbol{a}_{i^\star}\|^2|\kappa_{i^\star}^t|^2}{\beta} \geq 0.$$

This proves (18).

(b) Now, if $s_{i^\star}^t < 1$, according to (5) we have

$$r_{i^\star}^t = \frac{\left(G_{i^\star}(\boldsymbol{x}^t) + \mu_{i^\star}|\kappa_{i^\star}^t|^2/2\right)^2}{2(\mu_{i^\star} + \frac{\|\boldsymbol{a}_{i^\star}\|^2}{\beta})|\kappa_{i^\star}^t|^2}. \tag{19}$$

With $r_{i^\star}^t$ given by (19), (18) becomes

$$\frac{\left(G_{i^\star}(\boldsymbol{x}^t) + \mu_{i^\star}|\kappa_{i^\star}^t|^2/2\right)^2}{2(\mu_{i^\star} + \frac{\|\boldsymbol{a}_{i^\star}\|^2}{\beta})|\kappa_{i^\star}^t|^2} \geq G_{i^\star}(\boldsymbol{x}^t)\frac{\mu_{i^\star}}{\mu_{i^\star} + \frac{\|\boldsymbol{a}_{i^\star}\|^2}{\beta}},$$

and rearranging the items, it successively becomes

$$\frac{\left(G_{i^\star}(\boldsymbol{x}^t) + \mu_{i^\star}|\kappa_{i^\star}^t|^2/2\right)^2}{2|\kappa_{i^\star}^t|^2} \geq G_{i^\star}(\boldsymbol{x}^t)\mu_{i^\star}$$

$$\left(G_{i^\star}(\boldsymbol{x}^t) + \mu_{i^\star}|\kappa_{i^\star}^t|^2/2\right)^2 \geq 2G_{i^\star}(\boldsymbol{x}^t)|\kappa_{i^\star}^t|^2\mu_{i^\star}$$

$$G_{i^\star}(\boldsymbol{x}^t)^2 + (\mu_{i^\star}|\kappa_{i^\star}^t|^2/2)^2 - G_{i^\star}(\boldsymbol{x}^t)|\kappa_{i^\star}^t|^2\mu_{i^\star} \geq 0$$

$$\left(G_{i^\star}(\boldsymbol{x}^t) - \mu_{i^\star}|\kappa_{i^\star}^t|^2/2\right)^2 \geq 0,$$

which always holds and therefore recovers the claim, i.e., (18).

Hence in both cases (18) holds. Now, plugging (18) and $G(\boldsymbol{x}^t) \geq \epsilon(\boldsymbol{x}^t)$ in (4) yields

$$\epsilon(\boldsymbol{x}^{t+1}) - \epsilon(\boldsymbol{x}^t) = F(\boldsymbol{x}^{t+1}) - F(\boldsymbol{x}^t) \leq -r_{i^\star}^t$$

$$\leq -G(\boldsymbol{x}^t)\max_{i\in[d]}\frac{G_i(\boldsymbol{x}^t)\mu_i}{G(\boldsymbol{x}^t)\left(\mu_i + \frac{\|\boldsymbol{a}_i\|^2}{\beta}\right)} \leq -\epsilon(\boldsymbol{x}^t)\max_{i\in[d]}\frac{G_i(\boldsymbol{x}^t)\mu_i}{G(\boldsymbol{x}^t)\left(\mu_i + \frac{\|\boldsymbol{a}_i\|^2}{\beta}\right)}, \quad (20)$$

that results in

$$\epsilon(\boldsymbol{x}^{t+1}) \leq \epsilon(\boldsymbol{x}^t) - \epsilon(\boldsymbol{x}^t)\max_{i\in[d]}\frac{G_i(\boldsymbol{x}^t)\mu_i}{G(\boldsymbol{x}^t)\left(\mu_i + \frac{\|\boldsymbol{a}_i\|^2}{\beta}\right)}, \quad (21)$$

which gives

$$\epsilon(\boldsymbol{x}^{t+1}) \leq \epsilon(\boldsymbol{x}^t)\left(1 - \max_{i\in[d]}\frac{G_i(\boldsymbol{x}^t)\mu_i}{G(\boldsymbol{x}^t)\left(\mu_i + \frac{\|\boldsymbol{a}_i\|^2}{\beta}\right)}\right), \quad (22)$$

As (22) holds for all $t$, we conclude the proof. $\qquad\square$

**Proof:** **[Proof of Theorem 2]** Similar to the proof of Theorem 1, we prove the theorem for the following selection rule: At time $t$, the coordinate $i$ with the largest $G_i(\boldsymbol{x}^t)$ is chosen. Since the optimal selection rule for minimizing the bound in Lemma 1 is to select the coordinate $i$ with the largest $r_i^t$ in (4), as shown by Proposition 1, the convergence guarantees provided here holds for max_r as well.

The bound (7) is proven by using induction.

Suppose that (7) holds for some $t \geq t_0$. We want to verify it for $t + 1$. Let $i^\star = \text{argmax}_{i\in[d]}G_i(\boldsymbol{x}^t)$. We study two cases $s_{i^\star}^t = 1$ and $s_{i^\star}^t < 1$ separately, where $s_i^t$ is defined in (3) for $i \in [d]$.

(a) If $s_{i^\star}^t = 1$, then first we show that

$$\epsilon(\boldsymbol{x}^{t+1}) \leq \epsilon(\boldsymbol{x}^t) \cdot \left(1 - \frac{1}{2d}\right), \quad (23)$$

second we show that induction hypothesis (7) holds. Since $s_{i^\star}^t = 1$, (3) yields that

$$G_{i^\star}(\boldsymbol{x}^t) \geq \frac{|\kappa_{i^\star}^t|^2\|\boldsymbol{a}_{i^\star}\|^2}{\beta} + \frac{\mu_i|\kappa_{i^\star}^t|}{2},$$

that gives

$$G_{i^\star}(\boldsymbol{x}^t) \geq \frac{|\kappa_{i^\star}^t|^2\|\boldsymbol{a}_{i^\star}\|^2}{\beta},$$

which, combined with (5), implies that

$$r_{i^\star}^t = G_{i^\star}(\boldsymbol{x}^t) - \frac{|\kappa_{i^\star}^t|^2 \|\boldsymbol{a}_{i^\star}\|^2}{2\beta} \geq \frac{G_{i^\star}(\boldsymbol{x}^t)}{2}. \tag{24}$$

Using $F(x^{t+1}) - F(x^t) = \epsilon(\boldsymbol{x}^{t+1}) - \epsilon(\boldsymbol{x}^t)$ and (24), we can rewrite (4) as

$$\epsilon(\boldsymbol{x}^{t+1}) - \epsilon(\boldsymbol{x}^t) \leq -\frac{G_{i^\star}(\boldsymbol{x}^t)}{2}.$$

As $i^\star$ is the coordinate with the largest $G_i(\boldsymbol{x}^t)$, we have

$$\epsilon(\boldsymbol{x}^{t+1}) - \epsilon(\boldsymbol{x}^t) \leq -\frac{G_{i^\star}(\boldsymbol{x}^t)}{2} \leq -\frac{G(\boldsymbol{x}^t)}{2d}. \tag{25}$$

According to weak duality, $\epsilon(\boldsymbol{x}^t) \leq G(\boldsymbol{x}^t)$. Plugging this in (25) yields

$$\epsilon(\boldsymbol{x}^{t+1}) - \epsilon(\boldsymbol{x}^t) \leq -\frac{G(\boldsymbol{x}^t)}{2d} \leq -\frac{\epsilon(\boldsymbol{x}^t)}{2d}, \tag{26}$$

and therefore

$$\epsilon(\boldsymbol{x}^{t+1}) \leq \epsilon(\boldsymbol{x}^t) \cdot \left(1 - \frac{1}{2d}\right). \tag{27}$$

Now, by plugging (7) in (27) we prove the inductive step at time $l + 1$:

$$\epsilon(\boldsymbol{x}^{t+1}) \leq \frac{\frac{8L^2\eta^2}{\beta}}{2d + t - t_0}\left(1 - \frac{1}{2d}\right)$$

$$\leq \frac{\frac{8L^2\eta^2}{\beta}}{2d + t + 1 - t_0}.$$

(b) If $s_{i^\star}^t < 1$, the marginal decreases in (5) becomes

$$r_{i^\star}^t = \frac{\left(G_{i^\star}(\boldsymbol{x}^t) + \mu_{i^\star}\frac{|\kappa_{i^\star}^t|^2}{2}\right)^2}{2|\kappa_{i^\star}^t|^2\left(\mu_{i^\star} + \frac{\|\boldsymbol{a}_{i^\star}\|^2}{\beta}\right)}. \tag{28}$$

Next, we show that

$$r_{i^\star}^t \geq \frac{G_{i^\star}^2(\boldsymbol{x}^t)\beta}{2|\kappa_{i^\star}^t|^2\|\boldsymbol{a}_{i^\star}\|^2}. \tag{29}$$

To prove (29), we plug (28) in (29) and rearrange the terms which gives

$$\frac{\|\boldsymbol{a}_{i^\star}\|^2}{\beta}\left(\mu_{i^\star}^2\frac{|\kappa_{i^\star}^t|^4}{4} + G_{i^\star}(\boldsymbol{x}^t)\mu_{i^\star}|\kappa_{i^\star}^t|^2\right) \geq \mu_{i^\star}G_{i^\star}^2(\boldsymbol{x}^t), \tag{30}$$

(30) holds because of (3). More precisely, if we plug the value of $s_{i^\star}^t < 1$ in (3) we get

$$G_{i^\star}(\boldsymbol{x}^t) \leq \frac{|\kappa_{i^\star}^t|^2\|\boldsymbol{a}_{i^\star}\|^2}{\beta} + \frac{\mu_i|\kappa_{i^\star}^t|}{2}, \tag{31}$$

which shows the correctness of (30), hence (29).

According to Lemma 22 of [20] or Lemma 2.7 of [15] we know $|\kappa_i^t| \leq 2L$. Plugging $|\kappa_i^t| \leq 2L$ in (29) yields

$$r_{i^\star}^t \geq \frac{G_{i^\star}^2(\boldsymbol{x}^t)\beta}{8L^2\|\boldsymbol{a}_{i^\star}\|^2}.$$

Next, using weak duality and the definition of $\eta$ in Theorem 2, we lower bound $r_{i^\star}^t$ by

$$r_{i^\star}^t \geq \left(\frac{G_{i^\star}(\boldsymbol{x}^t)}{G(\boldsymbol{x}^t)\,\|\boldsymbol{a}_{i^\star}\|}\right)^2 \frac{G^2(\boldsymbol{x}^t)\beta}{8L^2}$$

$$\geq \frac{\epsilon^2(\boldsymbol{x}^t)\beta}{8L^2\eta^2}. \tag{32}$$

Hence we have

$$\epsilon(\boldsymbol{x}^{t+1}) - \epsilon(\boldsymbol{x}^t) \leq -r_{i^\star}^t \leq -\frac{\epsilon^2(\boldsymbol{x}^t)\beta}{8L^2\eta^2},$$

and therefore

$$\epsilon(\boldsymbol{x}^{t+1}) \leq \epsilon(\boldsymbol{x}^t)\left(1 - \frac{\epsilon(\boldsymbol{x}^t)\beta}{8L^2\eta^2}\right). \tag{33}$$

Let $f(y) = y\left(1 - {}^{y\beta}/_{8L^2\eta^2}\right)$, as $f'(y) > 0$ for $y < {}^{4L^2\eta^2}/_\beta$, plugging (7) in (33) yields

$$\epsilon(\boldsymbol{x}^t)\left(1 - \frac{\epsilon(\boldsymbol{x}^t)\beta}{8L^2\eta^2}\right) \leq \frac{\frac{8L^2\eta^2}{\beta}}{2d + t - t_0}\left(1 - \frac{\frac{8L^2\eta^2}{\beta}}{2d + t - t_0}\frac{\beta}{8L^2\eta^2}\right). \tag{34}$$

Now, we prove the inductive step at time $t + 1$ by using (34):

$$\epsilon(\boldsymbol{x}^{t+1}) \leq \frac{\frac{8L^2\eta^2}{\beta}}{2d + t - t_0} \cdot \left(1 - \frac{\frac{8L^2\eta^2}{\beta}}{2d + t - t_0}\frac{\beta}{8L^2\eta^2}\right)$$

$$\leq \frac{\frac{8L^2\eta^2}{\beta}}{2d + t + 1 - t_0}.$$

To conclude the proof, we need to show that the induction base case is correct, i.e., we need to show that

$$\epsilon(\boldsymbol{x}^{t_0}) \leq \frac{4L^2\eta^2}{\beta d}. \tag{35}$$

First, we rewrite (5) using $r_{i^\star}^t \geq {}^{\epsilon^2(\boldsymbol{x}^t)\beta}/_{8L^2\eta^2}$ for $s_{i^\star}^t < 1$ and $r_{i^\star}^t \geq {}^{\epsilon(\boldsymbol{x}^t)}/_{2d}$ for $s_{i^\star}^t = 1$ as

$$\epsilon(\boldsymbol{x}^{t+1}) - \epsilon(\boldsymbol{x}^t) \leq -r_{i^\star}^t \leq -\mathbb{1}\{s_{i^\star}^t = 1\}\frac{\epsilon(\boldsymbol{x}^t)}{2d} - \mathbb{1}\{s_{i^\star}^t < 1\}\frac{\epsilon^2(\boldsymbol{x}^t)\beta}{8L^2\eta^2}. \tag{36}$$

From (36), for $l < t_0$ we have

$$\epsilon(\boldsymbol{x}^{t+1}) \leq \epsilon(\boldsymbol{x}^t)\left(1 - \mathbb{1}\{s_{i^\star}^t = 1\}\frac{1}{2d} - \mathbb{1}\{s_{i^\star}^t < 1\}\frac{\epsilon(\boldsymbol{x}^t)\beta}{8L^2\eta^2}\right)$$

$$\leq \epsilon(\boldsymbol{x}^t)\left(1 - \min\left\{\frac{1}{2d}, \frac{\epsilon(\boldsymbol{x}^t)\beta}{8L^2\eta^2}\right\}\right)$$

$$\leq \epsilon(\boldsymbol{x}^t)\left(1 - \min\left\{\frac{1}{2d}, \frac{\epsilon(\boldsymbol{x}^{t_0})\beta}{8L^2\eta^2}\right\}\right), \tag{37}$$

where (37) holds because for $t \leq t_0$ we know that $\epsilon(\boldsymbol{x}^{t_0}) \leq \epsilon(\boldsymbol{x}^t)$. We use the proof by contradiction to check the induction base, i.e., we show that assuming $\epsilon(\boldsymbol{x}^{t_0}) > {}^{4L^2\eta^2}/_{\beta d}$ results in a contradiction. If $\epsilon(\boldsymbol{x}^{t_0}) > {}^{4L^2\eta^2}/_{\beta d}$, then

$$\frac{1}{2d} = \min\left\{\frac{1}{2d}, \frac{\epsilon(\boldsymbol{x}^{t_0})\beta}{8L^2\eta^2}\right\}. \tag{38}$$

From (37) and (38) we get

$$\epsilon(\boldsymbol{x}^{t_0}) \leq \epsilon(\boldsymbol{x}^0)\left(1 - \frac{1}{2d}\right)^{t_0}. \tag{39}$$

Using the inequality $1 + y < \exp(y)$ for $y < 1$ we have

$$\epsilon(\boldsymbol{x}^{t_0}) \leq \epsilon(\boldsymbol{x}^0) \exp(-\frac{t_0}{2d}) = \epsilon(\boldsymbol{x}^0) \exp(-\log \frac{d\beta\epsilon(\boldsymbol{x}^0)}{4L^2\eta^2})$$

$$= \epsilon(\boldsymbol{x}^0) \frac{4L^2\eta^2}{\beta d\epsilon(\boldsymbol{x}^0)} = \frac{4L^2\eta^2}{\beta d},$$

which shows that the induction base holds and this concludes the proof.

$\square$

**Proof:** [**Proof of Proposition 2**] The proof is similar to the proof of Theorem 2 and it uses induction. We highlight the differences here. To make the proof easier, we simplify the definition of $s_i^t$ in (3) and the marginal decrease $r_i^t$ in (5) by using the upper bound $|\kappa_i^t| \leq 2L$ (recall that $L = L_i$ for all $i$ in Proposition 2). The upper bound $|\kappa_i^t| \leq 2L$ follows from Lemma 22 of [20]. The starting point of the proof is the following equation

$$F(\boldsymbol{x}^{t+1}) \leq F(\boldsymbol{x}^t) - s_i^t G_i(\boldsymbol{x}^t) + 2 \frac{(s_i^t)^2 \|\boldsymbol{a}_1\|^2}{\beta} L^2, \tag{40}$$

which is derived by upper bounding (14) using $|\kappa_i^t| \leq 2L$ and $\mu_i = 0$ for all $i \in [d]$, which holds since $g_i(\cdot)$ are not strongly convex. Equation (40) holds for $s_i^t \in [0,1]$, and for all $i \in [d]$. After minimizing the right-hand side of (40) with respect to $s_i^t$, we attain the following *new* $s_i^t$ and the *new* marginal decrease $r_i^t$:

$$s_i^t = \min \left\{ 1, \frac{G_i^t}{4L^2\|\boldsymbol{a}_1\|^2/\beta} \right\}, \tag{41}$$

and

$$r_i^t = \begin{cases} G_i^t - \frac{2\|\boldsymbol{a}_1\|^2 L^2}{\beta} & \text{if } s_i^t = 1, \\ \frac{(G_i^t)^2}{8L^2\|\boldsymbol{a}_1\|^2/\beta} & \text{otherwise.} \end{cases} \tag{42}$$

Hereafter, let

$$\alpha = \frac{8L^2\|\boldsymbol{a}_1\|^2}{\beta \left( \varepsilon/d^2 + (1-\varepsilon)\eta^2/c \right)}$$

as defined in Proposition 2.

Now, suppose that (7) holds for some $t \geq t_0$. We want to verify it for $t+1$. We start the analysis by computing the expected marginal decrease for $\varepsilon$ in Algorithm 1,

$$\mathbb{E}\left[ r_i^t | \boldsymbol{x}^t \right] \geq \frac{\varepsilon}{d} \left( \sum_{s_i^t=1} r_i^t + \sum_{s_i^t<1} r_i^t \right) + (1-\varepsilon) \frac{r_{i^\star}^t}{c}, \tag{43}$$

where $c$ is a finite constant in Proposition 2 and $i^\star = \arg\max_{i \in [d]} r_i^t$. The expectation is with respect to the random choice of the algorithm.

When $s_i^t = 1$, from (41) we have $G_i^t \geq 4L^2\|\boldsymbol{a}_1\|^2/\beta$ and from (42) we have $r_i^t \geq 2L^2\|\boldsymbol{a}_1\|^2/\beta$. Plugging $r_i^t \geq 2L^2\|\boldsymbol{a}_1\|^2/\beta$ when $s_i^t = 1$ in (43) yields

$$\mathbb{E}\left[ r_i^t | \boldsymbol{x}^t \right] \geq \frac{\varepsilon}{d} \left( \sum_{s_i^t=1} \frac{2L^2\|\boldsymbol{a}_1\|^2}{\beta} + \sum_{s_i^t<1} \frac{G_i^2(\boldsymbol{x}^t)\beta}{8L^2\|\boldsymbol{a}_1\|^2} \right) + (1-\varepsilon)\frac{r_{i^\star}^t}{c}, \tag{44}$$

(a) If $s_{i^\star}^t = 1$, then the cost function decreases at least by

$$\mathbb{E}\left[ r_i^t | \boldsymbol{x}^t, s_{i^\star}(\boldsymbol{x}^t) = 1 \right] \geq \frac{2L^2\|\boldsymbol{a}_1\|^2}{\beta} \left( \frac{\varepsilon}{d} + \frac{1-\epsilon}{c} \right). \tag{45}$$

(b) Let $s_{i^\star}^t < 1$, from the definition of $s_{i^\star}^t$ we know that $G_i(\boldsymbol{x}^t) \le G_{i^\star}(\boldsymbol{x}^t)$ for all $i \in [d]$, hence we deduce that $s_i^t < 1$ for all $i \in [d]$, then (44) reads as

$$
\mathbb{E}\left[r_i^t | \boldsymbol{x}^t, s_{i^\star}(\boldsymbol{x}^t) < 1\right] \ge \frac{\varepsilon}{d}\left(\sum_{i=1}^d \frac{G_i^2(\boldsymbol{x}^t)\beta}{8L^2\|\boldsymbol{a}_1\|^2}\right) + (1-\varepsilon)\frac{G_{i^\star}^2(\boldsymbol{x}^t)\beta}{8L^2 c\|\boldsymbol{a}_1\|^2}
$$

$$
\ge \frac{\beta}{8L^2\|\boldsymbol{a}_1\|^2}\left(\varepsilon\frac{\left(\sum_{i=1}^d G_i(\boldsymbol{x}^t)\right)^2}{d^2} + (1-\varepsilon)\frac{G_{i^\star}^2(\boldsymbol{x}^t)}{c}\right)
$$

$$
\ge \frac{\beta}{8L^2\|\boldsymbol{a}_1\|^2}\left(\varepsilon\frac{G^2(\boldsymbol{x}^t)}{d^2} + (1-\varepsilon)\frac{G^2(\boldsymbol{x}^t)}{\eta^2 c}\right), \tag{46}
$$

where (46) follows from the assumption $G(\boldsymbol{x}^t) \le \eta G_{i^\star}(\boldsymbol{x}^t)$ in Proposition 2. Similar to the proof of Theorem 2, we plug the inequality $\epsilon(\boldsymbol{x}^t) < G(\boldsymbol{x}^t)$ in (46) and get

$$
\mathbb{E}\left[r_i^t | \boldsymbol{x}^t, s_{i^\star}(\boldsymbol{x}^t) < 1\right] \ge \frac{\beta\epsilon^2(\boldsymbol{x}^t)}{8L^2\|\boldsymbol{a}_1\|^2}\left(\frac{\varepsilon}{d^2} + \frac{(1-\varepsilon)}{\eta^2 c}\right) = \frac{\epsilon^2(\boldsymbol{x}^t)}{\alpha}. \tag{47}
$$

Next, we use (45), (47) and use the tower property to check the induction hypothesis

$$
\mathbb{E}[\epsilon(\boldsymbol{x}^{t+1})] - \mathbb{E}[\epsilon(\boldsymbol{x}^t)] \le \mathbb{E}\left[\mathbf{1}\{s_{i^\star}^t = 1\}\mathbb{E}\left[r_i^t | \boldsymbol{x}^t, s_{i^\star}^t = 1\right] + \mathbf{1}\{s_{i^\star}^t <= 1\}\mathbb{E}\left[r_i^t | \boldsymbol{x}^t, s_{i^\star}^t < 1\right]\right]
$$

$$
\le -\mathbb{E}\left[\mathbf{1}\{s_{i^\star}^t = 1\}\frac{2L^2\|\boldsymbol{a}_1\|^2}{\beta}\left(\frac{\varepsilon}{d} + \frac{1-\varepsilon}{c}\right) + \mathbf{1}\{s_{i^\star}^t < 1\}\frac{\epsilon^2(\boldsymbol{x}^t)}{\alpha}\right]. \tag{48}
$$

As we assumed

$$
\epsilon^2(\boldsymbol{x}^t) \le \epsilon^2(\boldsymbol{x}^0) \le \frac{2\alpha L^2\|\boldsymbol{a}_1\|^2}{\beta}\left(\frac{\varepsilon}{d} + \frac{1-\varepsilon}{c}\right)
$$

in Proposition 2, we have

$$
\min\left\{\frac{2L^2\|\boldsymbol{a}_1\|^2}{\beta}\left(\frac{\varepsilon}{d} + \frac{1-\varepsilon}{c}\right), \frac{\epsilon^2(\boldsymbol{x}^t)}{\alpha}\right\} = \frac{\epsilon^2(\boldsymbol{x}^t)}{\alpha}.
$$

Hence, (48) becomes

$$
\mathbb{E}[\epsilon(\boldsymbol{x}^{t+1})] - \mathbb{E}[\epsilon(\boldsymbol{x}^t)] \le -\mathbb{E}\left[\frac{\epsilon^2(\boldsymbol{x}^t)}{\alpha}\right] \le -\frac{\mathbb{E}[\epsilon(\boldsymbol{x}^t)]^2}{\alpha}, \tag{49}
$$

where the last inequality is because of the Jensen's inequality (i.e., $\mathbb{E}[\epsilon(\boldsymbol{x}^t)]^2 \le \mathbb{E}[\epsilon^2(\boldsymbol{x}^t)]$). By rearranging the terms in (49) we get

$$
\mathbb{E}[\epsilon(\boldsymbol{x}^{t+1})] \le \mathbb{E}\left[\epsilon(\boldsymbol{x}^t)\right]\left(1 - \frac{\mathbb{E}\left[\epsilon(\boldsymbol{x}^t)\right]}{\alpha}\right) \tag{50}
$$

Now, let $f(y) = y\left(1 - \frac{y}{\alpha}\right)$, as $f'(y) > 0$ for $y < \alpha/2$, we can plug (8) in (50) and prove the inductive step at time $t+1$;

$$
\mathbb{E}[\epsilon(\boldsymbol{x}^{t+1})] \le \mathbb{E}\left[\epsilon(\boldsymbol{x}^t)\right]\left(1 - \frac{\mathbb{E}\left[\epsilon(\boldsymbol{x}^t)\right]}{\alpha}\right)
$$

$$
\le \frac{\alpha}{2+t-t_0}\cdot\left(1 - \frac{1}{2+t-t_0}\right) \le \frac{\alpha}{2+t+1-t_0}. \tag{51}
$$

Finally, we need to show that the induction basis indeed is correct. By using the inequality (49) for $t = 1, \ldots, t_0$ we get

$$
\mathbb{E}[\epsilon(\boldsymbol{x}^{t_0})] \le \epsilon(\boldsymbol{x}^0) - \sum_{t=0}^{t_0-1}\frac{\mathbb{E}[\epsilon(\boldsymbol{x}^t)]^2}{\alpha}, \tag{52}
$$

Table 2: Statistics of the datasets. The first three datasets are used for regression and the last two for binary classification.

| | #classes | #datapoints | #features | %nonzero |
|---|---|---|---|---|
| usps | 10 | 7291 | 256 | 100% |
| aloi | 1000 | 108000 | 128 | 24% |
| protein | 3 | 17766 | 357 | 29% |
| w8a | 2 | 49749 | 300 | 4% |
| a9a | 2 | 32561 | 123 | 11% |

since at each iteration the cost function decreases, we have $\epsilon(\boldsymbol{x}^{t+1}) \leq \epsilon(\boldsymbol{x}^t)$ for all $t \geq 0$. Therefore, if $\mathbb{E}[\epsilon(\boldsymbol{x}^t)] \leq \alpha/2$ for any $0 \leq t \leq t_0$, we can conclude that $\mathbb{E}[\epsilon(\boldsymbol{x}^{t_0})] \leq \alpha/2$. We prove the induction hypothesis by showing that $\mathbb{E}[\epsilon(\boldsymbol{x}^{t_0})] > \alpha/2$ results in a contradiction. With this assumption, (52) becomes

$$\mathbb{E}[\epsilon(\boldsymbol{x}^{t_0})] \leq \epsilon(\boldsymbol{x}^0) - t_0\frac{\alpha}{4} = \epsilon(\boldsymbol{x}^0)\left(1 - t_0\frac{\alpha}{4\epsilon(\boldsymbol{x}^0)}\right), \tag{53}$$

Next, we use the inequality $1 + y \leq \exp(y)$ with (53)

$$\mathbb{E}[\epsilon(\boldsymbol{x}^{t_0})] \leq \epsilon(\boldsymbol{x}^0)\exp\left(-t_0\frac{\alpha}{4\epsilon(\boldsymbol{x}^0)}\right). \tag{54}$$

Plugging

$$t_0 = \frac{4\epsilon(\boldsymbol{x}^0)}{\alpha}\log(\frac{2\epsilon(\boldsymbol{x}^0)}{\alpha})$$

in (54) yields

$$\mathbb{E}[\epsilon(\boldsymbol{x}^{t_0})] \leq \frac{\alpha}{2}, \tag{55}$$

which proves the induction basis and concludes the proof.

$\square$