[Reviews · NeurIPS 2018]

Reviewer 1



The paper introduce a coordinate descent algorithm with an adaptive sampling à la Gauss-Southwell. Based on a descent lemma that quantifies the decay of the objective function when a coordinate is selected, the authors propose the "max_r" strategy that iteratively choose the coordinate that yields to the largest decrease. The paper follows recent developments on coordinate descent notably (Csiba et al 2015), (Nutini et al 2015), (Perekrestenko et al 2017) with an improved convergence bounds. As for previous adaptive sampling, the proposed method require a computational complexity equivalent to a full gradient descent which can be prohibitive in large scale optimization problem. To overcome this issue, the authors propose to learn the best coordinate by approximating the "max_r" strategy. The idea is to delay the updates of the duality gap (needed for max_r) along epochs and sample the coordinates by mixing uniform selection and steepest descent selection (with approximated marginal decrease). This attenuates the computational complexity while maintaining fast convergence. The proposed method is novel and simple, it will be beneficial to community of optimization for machine learning. It applies for a large class of problem and shows interesting direction for adaptive sampling techniques. In remark 2, showing that max_r coincide with Gauss-Southwell for (one particular example !) Ridge regularization is not sufficient to conclude that it is a "generalization". So this claims should be removed or properly justified. The proof of theorem 1 suggest that selecting the coordinate with largest marginal gap decrease is sufficient. Hence if max_r outperform ada_gap, then the complexity bound in theorem 1 is not tight and does not reflect the actual performance. In theorem 2, eps_p(x_0) is not needed here. In section 2.4, the mix of algorithm and main text is pretty unpleasant for reading. Should be nice to include more experiment of the effect of c(E, epsilon) at least in the appendix. In definition 3, the first inequality should be a strict inequality. There is a missing term "-eps_p(x^t)" in equation (21) ?

Reviewer 2



The paper introduces a new adaptive coordinate descent algorithm, which selects the coordinates based on a measurement of marginal decrease. The main idea is to measure the marginal decrease of each coordinate with the help of dual residues. The greedy algorithm consists of choosing the coordinate which provides the largest marginal decrease. However, the greedy version requires the information of all the coordinates which is computationally expensive. To overcome this issue, an adaptive variant which updates the information in an online style has been introduced. My major concern is about the theoretical analysis of the adaptive algorithm. In Proposition 2, the convergence rate of the algorithm is given under the assumption that the initial function gap f(x_0)-f* is smaller than some quantity Q. However, this assumption is unnatural and inappropriate due to the dependance between Q and f. It is not true that this condition will hold by scaling down the function f, because Q will also scale down. Moreover, Q depends on another highly non trivial quantity c, which depends on the functional property and probably the initial point. Thus, a more careful clarification is needed. Besides this negative point in the theoretical analysis, I find the experimental section unconvincing. In particular, the parameter choices seems very arbitrary, for example the regularization parameter. Moreover, it is not clear how the algorithm's key parameters are selected like \epsilon and E. For ridge regression, E is set to be n/2 and for other cases E is d/2. There might be a huge gap between the feature dimension d and the number of samples n. It sounds like taking the best value that works without any theoretical reason. Overall, the idea of using the marginal decrease as a guidance of coordinate selection is very interesting, but I do not support publication of the paper in the current state due to the weakness in both theoretical and the experimental results. Some additional remarks: 1) The l1 regularization is not L-bounded unless taking L=infinity 2) Some of the plots do not include any algorithm proposed in the paper, for example in logistic regression on dataset w8a or a9a in Figure 3. EDIT: I thank authors for addressing may concerns. Even though I still find the theoretical result a bit weak, I believe the paper provides interesting insight and ideas with outstanding empirical performance, as such, I raise my score accordingly.

Reviewer 3



Summary: -------- The paper introduces a general framework for doing coordinate descent using bandit algorithms. It provides compelling experimental results by comparing to similar algorithms. Quality and clarity: -------------------- The paper is very well written, from theory to experimental settings, with clear motivation and enough related literature. Proofs are readable and seem essentially correct (I haven't found errors by a quick read). Though not essential, it would ease reading if the authors explicit the results that they cite, in particular Lemma 22 of [22] does not immediately read their claim; it would also be nice to explicit the seminal inequality (14) on which the results are based (it is present in Lemma 3.1 of [16] but expliciting in the present framework would be nice). Originality and Significance: ----------------------------- The paper provides a generic framework to analyze coordinate descent using any bandit algorithm on generic residuals. The theoretical results pave the way to various implementations while the experimental shows its benefits on real data sets. The paper is really complete on both theoretical and experimental sides and it will be a good basis for other works to pursue. Minor comments: --------------- l 57: minimizing l 70: \bar x \in \argmin (not always unique), generally the authors always use =, a note should precise that it's a simplification Proposition 2: Replace \in O(d) by =O(d). In fact the authors use the notation \in O() several times l 166: Replace k==1 by K==1 l 496: form -> from l 461: L_i not L, then one gets r > and not = in the two following displayed equations Eq (45): In first inequality, oen misses a - and then it is not = but <=